# Isolation and Characterization of Copper- and Zinc- Binding Metallothioneins from the Marine Alga *Ulva compressa* (Chlorophyta)

**DOI:** 10.3390/ijms21010153

**Published:** 2019-12-25

**Authors:** Antonio Zúñiga, Daniel Laporte, Alberto González, Melissa Gómez, Claudio A. Sáez, Alejandra Moenne

**Affiliations:** 1Laboratory of Marine Biotechnology, Faculty of Chemistry and Biology, University of Santiago of Chile, Alameda 3363, Santiago 9170022, Chile; antonio.zt@gmail.com (A.Z.); alberto.gonzalezfi@usach.cl (A.G.); melissa.gomez@usach.cl (M.G.); 2HUB AMBIENTAL UPLA, Vicerrectoría de Investigación, Postgrado e Innovación, University of Playa Ancha, Avenida Carvallo 270, Valparaíso 2340000, Chile; claudio.saez@upla.cl; 3Laboratory of Aquatic Environmental Research, Center of Advances Studies, University of Playa Ancha, Traslaviña 450, Viña del Mar 2520000, Chile

**Keywords:** copper and zinc, expression in bacteria, metal accumulation, metallothioneins, marine alga, *Ulva compressa*

## Abstract

In this work, transcripts encoding three metallothioneins from *Ulva compressa* (UcMTs) were amplified: The 5′and 3′ UTRs by RACE-PCR, and the open reading frames (ORFs) by PCR. Transcripts encoding UcMT1.1 (*Crassostrea*-like), UcMT2 (*Mytilus*-like), and UcMT3 (*Dreissena*-like) showed a 5′UTR of 61, 71, and 65 nucleotides and a 3′UTR of 418, 235, and 193 nucleotides, respectively. UcMT1.1 ORF encodes a protein of 81 amino acids (MW 8.2 KDa) with 25 cysteines (29.4%), arranged as three motifs CC and nine motifs CXC; UcMT2 ORF encode a protein of 90 amino acids (9.05 kDa) with 27 cysteines (30%), arranged as three motifs CC, nine motifs CXC, and one motif CXXC; UcMT3 encode a protein of 139 amino acids (13.4 kDa) with 34 cysteines (24%), arranged as seven motifs CC and seven motifs CXC. UcMT1 and UcMT2 were more similar among each other, showing 60% similarity in amino acids; UcMT3 showed only 31% similarity with UcMT1 and UcMT2. In addition, UcMTs displayed structural similarity with MTs of marine invertebrates MTs and the terrestrial invertebrate *Caenorhabtidis elegans* MTs, but not with MTs from red or brown macroalgae. The ORFs fused with GST were expressed in bacteria allowing copper accumulation, mainly in MT1 and MT2, and zinc, in the case of the three MTs. Thus, the three MTs allowed copper and zinc accumulation in vivo. UcMTs may play a role in copper and zinc accumulation in *U. compressa*.

## 1. Introduction

Metallothioneins (MTs) are low molecular weight proteins, of around 10 kDa, that are rich in cysteine residues allowing the binding of divalent or monovalent metal ions such as Zn^2+^, Cd^2+^, Pb^2+^, Hg^2+^, Cu^1+^, Ag^1+^, among others [1,2,3]. MTs participate in metal accumulation and detoxification in vertebrates, invertebrates, plants, algae, and bacteria [1,2,3]. Cysteine residues in MTs are usually arranged as CC, CXC, and/or CXXC motifs and they correspond to around 30% of amino acids. In vertebrates, cysteine residues in MTs are contained in two domains, α and β, that are separated by a linker of variable sizes. Vertebrate MTs are rich in glycine and alanine amino acids, ranging from 10% to 20% residues [2] and invertebrate and plant MTs can contain histidine and aromatic residues [4,5]. MTs were first discovered in horse kidney, although they have been isolated from kidney and liver of other mammals [6]. In mammals, such as humans and mouse, there are four MT isoforms and the linker is constituted by three amino acids [2]. In fish, such as in rainbow trout, there are two MTs and the linker is formed by four amino acids [7]. In invertebrates, there are mainly two MTs, and the cysteine-rich domains are separated by a linker of four to seven amino acids and they have been described in organisms such as the nematode *Caenorhabtidis elegans* [8], the gastropod *Arianta arbustorum* [9], the snail *Helix pomatia* [10,11], the mollusks *Crassostrea virginica* [12], *Mytilus edulis* [13], and *Dreissena polymorpha* [14] and in the equinodern *Strongylocentrotus purpuratus* [15]. The yeast *Saccharomyces cerevisiae* displays two MTs, CUP-1 and CRS5, and the linker is constituted by three amino acids [16,17]. The cyanobacteria *Synechococcus* sp. showed a single MT, SmtA, and the linker that separates the two cysteine-rich domains is constituted by 15 amino acids [2]. The first MT isolated in plants was wheat germ MT, the first cloned MT in plants was *Mimulus guttatus MT*, and the first cloned MT from marine algae was *Fucus vesiculosus* MT [18,19,20]. Plant MTs can be classified in four types corresponding to type 1, 2, 3, and 4 [3]. The types 1, 2, and 3 MTs have cysteine-rich domains separated by a linker of around 40 amino acids and a linker of about 15 amino acids, and they differ in the arrangement of cysteines in cysteine-rich domains [3]. Type 4 MTs have mainly a linker of around 15 amino acids [3]. In *Arabidopsis thaliana,* there are seven MTs corresponding to MT1a, MT1c, MT2a, MT2b, MT3, MT4a, and MT4b and in *Populus trichcarpa x deltoides* there are six MTs corresponding to MT1a, MT1b, MT1a, MT1b, MT3a, and MT3b [21,22]. In addition, the study of algal genomes has shown that the brown macroalgae (Phaeophyceae) *Fucus serratus*, *Ectocarpus siliculosus*, and *Sargassum binderi*, and the red macroalgae (Rodophyceae) *Chondrus crispus* and *Euchema denticulatum* encode a single MT [3]. Red and brown algae MTs are related among each other considering the arrangement of cysteines [3]. Until now, no MTs have been cloned or characterized in green macroalgae (Chlorophyceae).

The green macroalga *Ulva compressa* is the dominant species in copper-polluted coastal areas of northern Chile and in other parts of the world [23,24]. It has been shown that the alga collected in the field accumulate copper in its tissue [23]. Furthermore, the alga cultivated in vitro with 2.5, 5, 7.5, and 10 µM for 0 to 12 days showed a linear accumulation of intracellular copper with increasing concentrations of the metal reaching a maximal accumulation of 620 µg g^−1^ of dry weight (DW), at day 12 with 10 µM copper [25]. In addition, *U. compressa* extrudes copper ions to the extracellular medium reaching a maximal concentration at intracellular level of around 900 µg g^−1^ of DW [24]. In contrast, the green alga *Ulva fasciata* cultivated with 0.3 µM copper for 14 days reached an intracellular level of copper of 2000 µg g^−1^ of DW suggesting that this alga may not extrude copper ions to the culture medium [26]. On the other hand, the level of MT transcripts in *U. compressa* cultivated with 7.5 and 10 µM of copper increased from days 3 to 12 [24]. Thus, it is possible that accumulation of intracellular copper is mediated by MTs in *U. compressa*. In this sense, it has been shown that *Arabidopsis thaliana* plants deficient in MT1a accumulate 30% less copper in the shoots than control plants [26] and *A. thaliana* mutants deleted in four MTs accumulate 45% less copper in the shoots [27]. In addition, rat fibroblasts having a deletion of MTI and MTII accumulate less copper than control cells and is subjected to an increase in oxidative stress [28]. Thus, it is possible that *U. compressa* can accumulate copper mediated by MTs. A transcriptomic analysis performed with the alga cultivated with 10 µM copper for 0 and 24 h allowed the identification of 7 potential MTs in *U. compressa* and their levels increased from days 3 to 12 of metal exposure [29].

In this work, we cloned three putative *U. compressa* MTs previously described as *Crassostrea*-like, *Mytilus*-like, and *Dreissena*-like MTs [26], which were renamed UcMT1, UcMT2, and UcMT3, respectively. In this work, the ORF of each UcMT was cloned and expressed in the bacteria *E. coli* as a fusion protein with a glutahione-S-transferase (GST) allowing accumulation of copper and zinc in vivo. Thus, the marine alga *U. compressa* may accumulate intracellular copper and zinc through UcMTs.

## 2. Results

### 2.1. Sequences of Transcripts Encoding UcMTs

Total RNA and mRNAs were isolated from *U. compressa* cultivated with 10 µM copper for three days. The 5′and 3′untranslated regions (UTR) of transcripts encoding three UcMTs were amplified using RACE-PCR technique, as well as the open reading frame (ORF) using conventional PCR. UcMT1.1 transcript (formerly *Crassostrea*-like *mt*) and protein are described in Table 1 (Figure 1). It is important to mention that two other transcripts that were closely related with UcMT1.1 were isolated, corresponding to UcMT1.2 and UcMT1.3; UcMT1.2 showed a deletion of 48 nucleotides after G in position 471 of the 3´UTR region of UcMT1.1, and UcMT1.3 displayed the same deletion mentioned before and a deletion of 58 nucleotides after C in position 607 in the 3´UTR of UcMT1.1 (Figure 1). UcMT2 transcript (formerly *Mytilus*-like *mt*) and protein is described in Table 1 (Figure 2) as well as UcMT3 transcript (formerly *Dreissena*-like *mt*) and protein (Figure 3). The linker region of UcMT1.1, UcMT2, and UcMT2 correspond to, 9, 9, and 23 amino acids, respectively (Figure 1, Figure 2 and Figure 3).

### 2.2. Similarities in Amino Acids of UcMTs and Hierarchical Clustering of MTs

UcMT1.1 and UcMT2 were more closely related among each other and shared 52.2% identity and 60% similarity in amino acids; indeed, both sequences contained arrangement of cysteines corresponding to 3 CC and 9 CXC motifs, but UcMT2 also contained a CXXC motif (Figure 4A). In contrast, UcMT3 shared only 30% similarity in amino acids with MT1.1 and MT2 and contained arrangements of cysteines corresponding to 7 CC and 7 CXC (Figure 4A). UcMT3 showed an extra N-terminal sequence of 10 amino acids, an internal additional sequence of 19 amino acids, and an extra C-terminal sequence of 8 amino acids, compared with UcMT1.1 and UcMT2 (Figure 4A). Thus, UcMT1 and UcMT2 may have derived from UcMT3 by deletions of initial, internal, and terminal nucleotide sequences.

The hierarchical clustering of vertebrate, invertebrate, and plants MTs constructed with 237 protein sequences (including the 3 UcMTs) demonstrated that UcMT1.1 and UcMT2 grouped mainly with marine crustacean MTs, such as those of the lobster *Homarus americanus* and the crabs *Carcinus maena* and *Scylla serrata* (Appendix A). In addition, UcMT3 clustered with the nematode *C. elegans* MTs, as well as with MTs of marine equinoderms MTs such as those of *Sterechinus neumayeri*, *Strongylocentrus purpuratus,* and *Paracentrotus livudus* (Appendix A). On the other hand, UcMTs grouped as a different clade with MTs from Rodophyceae and Phaeophyceae (Figure 4B). Thus, *U. compressa* MTs are more closely related with marine invertebrate MTs and the terrestrial invertebrate *C. elegans* MTs, and not to MTs from other marine macroalgae.

### 2.3. Expression of UcMTs-GST in Bacteria and Detection of GST-Tag

The ORFs of UcMTs were cloned in an *E. coli* expression vector, which allows the expression of MTs fused with the enzyme glutathione-S-transferase (GST) from the platyhelminthe *Schistosoma japonicum* (26 kDa), an enzyme that contain a single cysteine in the N-terminal domain, and do not bind metals. After 1 to 12 h of culture, the induction of protein expression with IPTG allows the visualization of increasing levels of UcMT1.1-GST (34.2 kDa. Figure 5A), UcMT2-GST (35.05 KDa, Figure 5B), and UcMT3-GST (39.4 KDa, Figure 5C). These proteins were detected by Western blot using an antibody prepared against *S. japonicum* GST indicating that the overexpressed proteins correspond to UcMTs fused with GST (Figure 5C).

### 2.4. UcMTs-GST-Mediated Accumulation of Copper or Zinc In Vivo

Transformed bacteria expressing MTs-GST were cultivated with 0.5 mM IPTG for 30 min, and with 1 mM copper and IPTG for 6 h. Control bacteria expressing only GST accumulate 0.35 µg of copper mg^−1^ of dry weight (DW), whereas those expressing UcMT1.1-GST, UcMT2-GST, and UcMT3-GST accumulate 1.8, 1.7, and 1.4 times more copper than the control, respectively (Figure 6A). On the other hand, transformed bacteria were cultivated with 0.5 mM IPTG for 30 min and, with 1 mM zinc and IPTG for 6 h. Control bacteria accumulate 0.49 µg mg^−1^ of zinc mg^−1^ of DW whereas those expressing UcMT1.1-GST, UcMT2-GST, and UcMT3-GST accumulate 4,1, 3.8, and 3.4 times more zinc than the control, respectively; although these increases were not significantly different among each other (Figure 6A). Thus, expression of UcMTs-GST mediates accumulation of copper and zinc in vivo.

## 3. Discussion

In this work, we isolated the complete sequence of transcripts encoding three MTs from the marine alga *U. compressa*. These transcripts encode UcMT1.1, UcMT2, and UcMT3 with a MW of 8.3, 9.05, and 13.4 kDa, respectively, and containing 24–30% of cysteines. It is important to mention that UcMT1.1 and UcMT2 seemed to be structurally more related among each other than to UcMT3. UcMT1.1 and UcMT2 showed deletions in amino acids sequences compared with MT3 at the initial, internal, and terminal part of the protein compared with UcMT3. Thus, UcMT2 and UcMT3 genes may have derived from UcMT3 gene. In the case of UcMT1, it seems that more than a single copy exists in *U. compressa* genome since three different transcripts of UcMT1 were isolated corresponding to UcMT1.1, UcMT1.2 and UcMT1.3. In this sense, it has been shown that the increase in number of copies of Cup1 MT gene in the yeast *S. cereviciae* allowed an increased tolerance to copper [30]. In addition, several copies of domains α and β of MTs exist in the mollusk *C. gigas* [12]. Thus, it is not surprising that UcMT1 may be present in multiple copies in *U. compressa* genome, but the latter remained to be confirmed. The sequencing of *U. compressa* genome is already in course.

Interestingly, UcMTs resemble marine invertebrate MTs. Mollusk MTs are constituted by 73–75 amino acids and contain 20–21 cysteine residues (28%), arranged mainly as CXC and CC motifs [5]. UcMTs are longer than mollusk MTs since they are constituted by 81, 90, and 139 amino acids; although the content of cysteines is similar to mollusk MTs (28–30%). However, mollusk MTs reported until now do not contain histidine or aromatic residues [5]. In addition, MT from marine crustacean and equinoderms showed cysteines (29–30%) arranged as CC and CXC and they do not contain histidine or aromatic aminoacids [25,26]. It is important to mention that we showed that UcMT1.1 contains a tyrosine, UcMT2 a histidine, and UcMT3 a tryptophan. In this sense, the two MTs found in the worm *C. elegans* are constituted by 75 amino acids and contain 19 cysteines (25%), arranged as CXC and CC, and contain a tyrosine and histidine residues [8]. Thus, UcMTs are more similar to *C. elegans* MTs but longer and contain a higher percentage of cysteines (28–30%). This indicates that UcMTs are unique MTs, and their sequences are more closely related with MTs of marine invertebrates and *C. elegans* MTs. It is not surprising that UcMTs showed similarities to marine invertebrate MTs since *F. vesiculosus* MT is also more related to mollusk MTs than to plant MTs [20]. *F. vesiculosus* MT showed a linker of 14 amino acids, which is longer than the linker of vertebrates MTs (4 aa) and the linkers of plant MTs (4 and 40 aa) [20]. The linker of UcMTs has a size of 9–13 amino acids that is more closely related with *F. vesiculosus* and invertebrate MTs linker size.

In addition, the sequence of UcMTs is different from the MT cloned from *F. vesiculosus*, which is constituted by 67 amino acids (6.9 kDa) and contains 16 cysteine residues (24%) arranged as CXC, but not as CC [3]. The only structural similarity of *Fucus mt* is that this transcript showed a short 5′UTR of 64 nucleotides and a long 3′UTR of 960 nucleotides [8]. Furthermore, MTs identified in the genomes of Rodophyceae *C. cripus* and *E. denticulatum* are constituted by 69 to 71 amino acids and contain 12 to 14 cysteines (20%) arranged mainly as CXC, but not as CC [2]. Furthermore, UcMTs grouped in a different clade compared with MTs from other marine macroalgae. Thus, UcMTs of the green macroalga *U. compressa* are distantly related with those found in red and brown macroalgae.

On the other hand, we showed that UcMTs-GST mediates the accumulation of copper and zinc in bacteria. In particular, UcMT1.1 and UcMT2, which are structurally more closely related, allowed higher accumulation of copper compared with UcMT3. Moreover, the three MTs allowed accumulation of zinc with similar efficiencies among them. Thus, the three UcMTs differentially bind copper and similarly bind zinc in vivo but their affinity for copper and zinc need to be further investigated. In this sense, it has been demonstrated that MTs are either more Zn- or Cu-binding thioneins, preference associated with cysteine arrangements and the nature of the other amino acids that constitute the MT [4,31]. Likewise, mouse MT1, MT2, and MT3 are more Zn-thionein and, in contrast, yeast cup-1, *Drosophila* MntA and MntB, and mouse MT4, are more Cu-thioneins [4,31]. Thus, the nature of UcMTs regarding their affinity for copper or zinc need to be further investigated.

It is now clearly established that green and red macroalgae are more closely related among each other, and with terrestrial plants, than with brown macroalgae [32,33,34]. Green and red macroalgae belong to the kingdom Plantae, as terrestrial plants [35] whereas brown macroalgae belong to the kingdom Chromalveolta [34]. The latter is based on the observations that green and red algae contain a plastid that derive from a single event of endosymbiosis by a cyanobacteria, whereas brown algae plastids derive from a secondary or tertiary endosymbiosis event of green or red microalgae [32,33]. Thus, it is unexpected that the three UcMTs of the green macroalga *U. compressa* are not closely related with other marine alga MTs, in particular to red macroalgae MTs. In contrast, the major similarity is with MTs of marine invertebrate and *C. elegans* MTs. Considering that marine algae appeared on earth around a billion years ago, and marine invertebrates emerged around 500 million years ago [34,36] and, moreover, that UcMTs are longer than MTs of marine invertebrates, it is then possible that marine invertebrates have acquired MTs genes from marine green macroalgae by horizontal gene transfer; however, this hypothesis need to be further investigated. Furthermore, it is possible that *U. compressa* contain additional MTs that differentially bind copper, zinc, and other heavy metals, as it has been predicted in [23], but the latter need to be further analyzed. In this sense, it has been shown that the equinoderm *Paracentrotus livudus* exhibits 7 MTs [35] suggesting that the four other potential UcMTs can be functional MTs. Thus, additional UcMTs may exist in the genome of *U. compressa* and these MTs will be cloned and characterized in the future.

In conclusion, we showed that transcripts encoding three MTs were cloned and sequenced; they encode unique MTs with homology with marine invertebrate and *C. elegans* MTs. UcMTs expressed in bacteria allowed copper and zinc accumulation in vivo. Thus, it is likely that these UcMTs may participate in copper and zinc accumulation in the marine alga *U. compressa*.

## 4. Materials and Methods

### 4.1. Sampling of Algae and Water Collection

The green macroalga *U. compressa* was collected during spring 2017 from the high intertidal zone at Cachagüa Beach (32°34′ S), a site in central Chile with no history of metal pollution [23]. Algal samples were transported to the laboratory in sealed plastic bags inside a cool-box. In the laboratory, material was rinsed three times with filtered seawater, cleaned manually, and sonicated three times for 2 min using an ultrasound bath (HiLab Innovation Systems, model SK221OHP) to remove epiphytes. The algae were maintained in aerated seawater under an irradiance of 50 μmoles m^−2^ s^−1^ on a photoperiod of 12 h:12 h light:dark cycle, at 14 °C for 4 days, prior to experimentation. Seawater was obtained from Quintay (33°12′ S), a pristine site in central Chile, filtered through 0.45 and 0.2 µm pore size membranes and stored in darkness at 4 °C.

### 4.2. Algal Culture and RNA Extraction

*U. compressa* (1 g of FT) was cultivated in 100 mL of filtrated seawater containing 10 µM CuCl_2_ (Merck, Darmstat, Germany) for 3 days. The alga was washed with 10 mM Tris-50 mM EDTA pH 7.0, in order to eliminate copper and other metals bound to cell walls [37].

### 4.3. Purification of Total RNAs and mRNAs for RACE-PCR

Total RNAs were extracted as described in [38]. *U. compressa* (150 mg of FT) was frozen in liquid nitrogen and pulverized in a mortar. One mL Trizol reagent (Invitrogen, Carlsbad, CA, USA) was added and the alga was homogenized with a pestle until thawing. The mixture was centrifuged at 12,000× *g* for 10 min at 4 °C, and the supernatant was recovered. Chloroform (200 µL) was added and the mixture was vortexed for 10 s and left at room temperature for 3 min. The mixture was centrifuged at 12,000× *g* for 15 min at 4 °C, and the aqueous phase was recovered. Isopropanol (500 µL) was added and the solution incubated for 10 min at room temperature. The solution was centrifuged at 12,000× *g* for 10 min at 4 °C, and the supernatant removed. The pellet was washed twice with 1 mL of 75% ethanol, gently vortexed, and centrifuged at 7000× *g* for 5 min at 4 °C. The ethanol phase was removed, the pellet dried for 15 min at room temperature, dissolved in 50 µL of ultrapure water treated with DEPC (water-DEPC), and incubated for 10 min at 60 °C.

Total RNAs were quantified using Nanodrop spectrophotometer (Tecan, Zürich, Switzerland); the integrity was verified by agarose gel electrophoresis and stored at −80 °C. Messenger RNAs were purified from 100 µg of total RNA using NucleoTrap mRNA minikit (Macherey-Nagel, Düren, Germany), mRNAs were eluted in 25 µL of water-DEPC, and normally 1 µg of mRNAs was obtained from 100 µg of total RNAs.

### 4.4. Amplification of 5′RACE cDNAs

The amplification of cDNAs was performed using MMLV reverse transcriptase kit (Promega, Madison, WI, USA). To this end, 4.1 µL of purified mRNAs (150 ng) were mixed with 2.5 µL of primer 1 for MT1 (5′GTAGCAGGCACAGTCGTCA3′), primer 2 for MT2 (5′AGGCCTAACAAGCAGCGTCC3′), or primer 3 for MT3 (5′CCGAGAGCGTGTCCTTACTT3′) at 10 µM, and 8.4 µL of water-DEPC were added to complete a final volume of 15 µL. The mixture was denatured at 65 °C for 5 min and cooled on ice for 1 min. Then, 0.7 µL of RNase inhibitor (10 U µL^−1^), 5 µL of MMLV buffer 5×, 1.5 µL of dNTPs mixture (10 mM of each dNTP), 1 µL de MMLV reverse transcriptase (200 U µL^−1^), and 1.8 µL of water- DEPC were added to complete a final volume of 25 µL. The mixture was sequentially incubated at 42 °C for 50 min, at 50 °C for 10 min, at 55 °C for 5 min, and at 70 °C for 10 min in order to inactivate MMLV reverse transcriptase. cDNAs were treated with RNAse H, purified using GEL/PCR purification mini kit (Favorgen Biotech Corp., Changzhi, Taiwan), and eluted in 40 µL of water-DEPC.

Purified cDNAs (14.5 µL) were incubated with 0.5 µL of terminal transferase (TdT), 20 U µL^−1^, New England Biolabs, Ipswich, MA, USA), 1 µL of dCTP, 2 µL of CoCl_2_ (2.5 mM), and 2 µL of TdT buffer 10×, to complete a final volume of 20 µL. The mixture was incubated at 37 °C, and at 75 °C for 20 min to inactivate TdT in order to obtain cDNAs having a 3′ C-tail.

The first round of amplification was performed using PCR kit (Favorgene, London, UK) and 1 µL of C-tailed cDNAs mixed with 10 µL of PCR mixture, 0.6 µL of 5′RACE adapter primer (5′GGCCACGCGTCGACTAGTACGGGIIGGGIIGGGIIG3′), and with 0.6 µL of primer 4 for MT1 (5′GGCATACGCACGTCTCGGG3′), primer 5 for MT2 (5′CTGCGTAACGACATAGCCGA3′), or primer 6 for MT3 (5′GCAGCCAGAATCGCAACTAC3′), at 10 µM, and 6.8 µL of water-DEPC to complete a final volume of 20 µL. The mixture was incubated at 95 °C for 3 min and subjected to 40 cycles of denaturation at 95 °C for 5 s, hybridization at 63 °C for 10 s and amplification at 72 °C for 15 s, using a real-time thermocycler RotorGene 6000. The second round of amplification was performed with the mixture diluted 100 times in distilled water-DEPC, and 14.4 µL of the cDNAs were mixed with 10 µL of PCR mixture, 0.6 µL of abridged universal adaptor primer (AUAP) (5′GGCCACGCGTCGACTAGTAC3′) and primer 7 for MT1, (5′GCAACCATCTTCGGTTTGGC3′), primer 8 for MT2, (5′ATCCTTCGCGGGTGAGCAAG3′), and primer 9 for MT3 (5′CACAGTTGCATTCTGCGGTT3′) at 10 µM and with 6.8 µL of water-DEPC to complete a final volume of 20 µL. The amplification was performed using 40 cycles of amplification mentioned before. The amplified fragments were analyzed in a 2% agarose gel, stained with SYBR green (Invitrogen, Carlsbad, CA, USA), and visualized on a UV trans-illuminator.

### 4.5. Amplification of 3′RACE cDNAs

Initial cDNAs were obtained using MMLV reverse transcriptase kit (Promega, Madison, WI, USA). To this end, 6.8 µL of mRNAs (250 ng) were mixed with 5 µL of 3′RACE adapter primer with an oligo-dT tail (5′GGCCACGCGTCGACTAGTACTTTTTTTTTTTTTTTTT3′) at 10 µM and with 3.2 µL of water-DEPC to complete a final volume of 15 µL. The mixture was denatured at 65 °C for 5 min and then cooled on ice for 1 min. Then, 0.7 µL of RNAse inhibitor (10 U µL^−1^), 5 µL of MMLV buffer 5×, 1.5 µL of dNTPs mixture (10 mM of each dNTP), 1 µL de MMLV reverse transcriptase (200 U µL^−1^), and 1.8 µL of water- DEPC were added to complete a final volume of 25 µL. The mixture was incubated at 42 °C for 1 h and at 70 °C for 10 min to inactivate MMLV reverse transcriptase and diluted to 50 µL with water-DEPC. cDNAs were treated with RNAse H, purified using GEL/PCR purification mini kit (Favorgen Biotech Corp., Changzhi, Taiwan), and eluted in 40 µL of water-DEPC.

The first round of amplification was performed using PCR kit (Favorgene, London, UK) and 1 µL of cDNAs were mixed with 0.6 µL of AUAP (5′GGCCACGCGTCGACTAGTAC3′), 0.6 µL of primer 10 for MT1 (5′CAGTGCCAAACCGAAGATGG3′), primer 11 for MT2 (5′GATGAGGGCTGTCCTTGCTC3′), or primer 12 for MT3 (5′AGTGTGATGCTGAGTGCTGT3′) at 10 µM and 6.8 µL of water-DEPC to complete a final volume of 20 µL. The mixture was incubated at 95 °C for 3 min and subjected to 40 cycles of denaturation at 95 °C for 5 s, hybridization at 63 °C for 10 s, and amplification at 72 °C for 15 s, using a real-time thermocycler RotorGene 6000. The second round of amplification was performed with the mixture diluted 100 times in distilled water-DEPC, and 14.4 µL of the cDNAs were mixed with 10 µL of PCR mixture, 0.6 µL of AUAP forward (5′GGCCACGCGTCGACTAGTAC3′) and with primer 13 for MT1 (5′GGTTGCAAGTGCTAGCTGAC3′), primer 14 for MT2 (5′GCTTGTTAGGCCTCAGTGGT3′), or primer 15 for MT3 (5′TGTCAGTGCGACAGCCTAA3′) and with 6.8 µL of water-DEPC to complete a final volume of 20 µL. The amplification was performed using 40 cycles of amplification mentioned before. The amplified fragments were analyzed in a 2% agarose gel, stained with SYBR green (Invitrogen, Carlsbad, CA, USA) and visualized on a UV trans-illuminator.

### 4.6. Amplification of UcMT ORFs

MT1.1 ORF was amplified using primer 16 forward (5′ATGGACTGCCGTTGCG3′) and primer 17 reverse (5′GCACTTGCAACCGCCAGAGC3′); MT2 ORF was amplified using primer 18 forward (5′ATGAACTGCTGTTGCGA3′) and primer 19 reverse (5′GACACAGCCCGGACAGGC3′); and MT3 ORF was amplified using primer 20 forward (5′ATGTCGTCTTGTTGTGAAGC3′) and primer 21 reverse (5′GGCTGTCGCACTGACACAG3′). The mixture was incubated at 95 °C for 2 min and subjected to 35 cycles of denaturation at 95 °C for 15 s, hybridization at 65 °C for UcMT1.1, 63 °C for MT2 and 62 °C for UcMT3, during 10 s, and amplification at 72 °C for 15 s, using a real-time thermocycler RotorGene 6000.

### 4.7. Cloning of UcMTs 5′and 3′UTRs, and UcMT ORFs in pGEM-T Vector

The 5′RACE and 3′RACE amplification fragments obtained in the second PCR (see above) and those of UcMTs ORFs were subjected to electrophoresis in 2% agarose gel. The piece of agarose gel containing the stained fragments was removed from the gel and placed in Eppendorf tubes. The amplified fragments were eluted from agarose using Gel/PCR purification kit (Favorgen, London, UK), recovered in 50 µL of water-DEPC and stored at 4 °C. Amplified fragments were ligated with the cloning vector pGEM-T easy (Promega, Madison, WI, USA) and transformed in *E. coli* competent cells One Shot TOP 10 (Invitrogen, Carlsbad, CA, USA). Transformed *E. coli* cells were cultivated in 10 mL of LB medium (10 g tryptone, 5 g yeast extract, and 100 g NaCl in 1 L of distilled water) supplemented with 100 µg mL^−1^ of ampicillin. The culture was centrifuged at 3,000 x g for 5 min in a centrifuge model Nuwind (Nuaire, Plymouth, MN, USA). Transformed pGEM-T vectors were purified from the bacterial pellet using Wizard Plus SV Miniprep DNA Purification System (Promega, Madison, WI, USA). To check cloning of 5′UTR fragments in pGEM-T, primers AUAP and primers 7, 8, and 9 were used. To check cloning of 3′UTR in pGEM-T, primer AUAP and primers 13, 14, and 15 were used. PCR conditions were identical to those used for the amplification of 5′and 3′RACE ends (mentioned above). To amplify the ORFs, primers 16 and 17 for MT1, primers 18 and 19 for MT2, and primers 20 and 21 for MT3 were used and to verify the insertion PCR conditions were those used to amplify ORFs (mentioned above). Cloned fragments were sequenced using an ABI3730XL (Macrogen, Seoul, Korea).

### 4.8. Cloning of UcMTs ORFS in pGEX Expression Vector

*UcMTs* were synthesized and subjected to codon optimization for expression in *E. coli* by Genscript (Piscataway, NJ, USA) and then ligated to the expression vector pGEX-5X-1 (Genscript) which allowed fusion of UcMTs ORFS with the enzyme glutahione-S-transferase (GST) from the platyhelminthes *Schistosoma japonicum* (26 Kda), an enzyme containing a single cysteine that does not bind metals; the fusion proteins were named UcMT1.1-GST, UcMT2-GST, and UcMT3-GST. The recombinant vectors were sequenced by Genscript to verify the correct insertion of complete ORFs.

### 4.9. Transformation of Expression Vectors in Bacteria

The recombinant expression vectors were transformed in competent *E. coli* strain BL21 (DE3) (Sigma-Aldrich, Saint Louis, MO, USA). To this end, 200 µL of competent cells BL21 (DE3) were incubated with 50 ng of recombinant expression vector containing UcMTs-GST. Then, 800 µL of SOC medium (2% (*w*/*v*) tryptone, 0.5% (*w*/*v*) of yeast extract, 10 mM NaCl, 2.5 mM KCl, 10 mM MgCl_2_, 10 mm MgSO_4_, and 20 mM glucose) were added, and cultivated at 37 °C for 45 min. An aliquot of 200 µL was cultured on solid LB medium containing 100 µg L^−1^ of carbenicillin in a Petri dish overnight. Individual colonies were selected for each UcMT, cultivated with LB medium, and stored at −80 °C in LB medium containing 15% glycerol.

### 4.10. Expression of UcMTs-GST in Bacteria

Recombinant *E. coli* were cultured 15 mL of LB medium until OD_600_= 0.6, 0.5 mM of isopropyl-β-D-1thiogalactopyranoside (IPTG) was added, and samples of 1 mL were obtained after 1, 3, 6, 9, and 12 h of culture at 37 °C. The samples were centrifuged at 7000× *g* for 10 min, washed with PBS pH 7.4 (10 mM Na_2_HPO_4_, 1.8 mM KH_2_PO_4_, 140 mM NaCl, 2.7 mM KCl), and centrifuged again in similar conditions. Pellets were suspended in 50 µL of protein loading buffer 2× (125 mM Tris HCl pH=6.8, 4% (*w*/*v*) SDS, 20% (*v*/*v*) glycerol, 10% (*v*/*v*) β-mercaptoethanol, 0.004% (*w*/*v*) bromophenol blue), and heated at 95 °C for 5 min. A sample of 10 µL was analyzed in a denaturant polyacrylamide gel (12%) and proteins were stained with Coomassie blue staining solution (25% (*v*/*v*) methanol, 5% (*v*/*v*) acetic acid, and 0.1% (*w*/*v*) of Comassie blue G-250).

### 4.11. Purification of UcMTs-GST by Affinity Column

A sample of 2 mL of *E. coli* transformed with expression vectors containing a UcMT-GST were added to 100 mL LB medium containing 100 µg mL^−1^ ampicillin and cultured at 37 °C overnight. A sample of 10 mL was added to 1 L of LB medium containing 100 µg mL^−1^ ampicillin and 0.5 mM IPTG, in quadruplicate, until OD_600_= 0.6 (aprox. 2.5 h). The 4 L of culture were centrifuged at 6000× *g* for 10 min. The pellet was washed twice with 300 mL of PBS and bacteria were suspended in 10 mL of PBS containing 5 mM dithiotreitol (DTT) and 1 tablet of protease inhibitor cocktail (Roche, Manheim, Germany). The bacterial suspension was sonicated for 20 s, with 20 s of pause, for 5 min. The suspension was centrifuged at 6000 rpm for 10 min and supernatant was recovered. Protein concentration was determined using Bradford method [39] and adjusted with PBS-5 mM DTT (PBS-DTT) to 1 mg mL^−1^. UcMTs-GST were purified by HPLC using GSTrap HP (General Electric, Uppsala, Sweden) at 5 bars of pressure, washed with PBS-DTT, and eluted with 3 mL of buffer 50 mL Tris-HCl-10 mM GSH. Normally, 1–2 mg of purified UcMT-GST was obtained from 4 L of bacterial culture and proteins were quantified using Bradford method [40].

### 4.12. Detection of UcMTs-GST with Anti-GST Antibody

Transformed bacteria were cultured in 100 mL of LB medium until OD_600_ = 0.6, 0.5 mM IPTG was added and the mixture incubated for 6 h. The culture was centrifuged at 6000× *g* for 5 min, the pellet suspended in 5 mL of buffer PBS, and sonicated for 20 s, with 20 s of pause, for 5 min. Proteins (20 µg) were separated by electrophoresis in a denaturant 12% polyacrylamide gel and transferred to a nitrocellulose membrane for 10 min using Trans Blot Turbo apparatus (BioRad, Hercules, CA, USA). The membrane was stained with Ponceau Red dye and washed with 10 mL of distilled water. The membrane was incubated in 10 mL TTBS (20 mM Tris-HCl pH 7.5, 0.1 mM NaCl, 0.05% Tween-20) containing 5% skim milk for 1 h, washed twice with 10 mL TTBS for 15 min, incubated with 10 mL TTBS containing 3% skim milk and the antibody anti-GST (Sigma-Aldrich, St Louis, MO, USA) diluted 5000 times, and washed four times in TTBS for 15 min. The membrane was incubated in TTBS containing 3% skim milk and the secondary antibody prepared against rabbit IgG coupled to hoseradish peroxidase (Agrisera, Vännas, Sweden) diluted 2000 times, for 1 h, and washed four times with TTBS for 15 min. Proteins were detected using ECL Western Blotting System kit (Amersham, Buckinghamshire, UK) and revealed using a C-Digits chemiluminiscence Western blot scanner Li-Cor (Lincoln, NE, USA) and Image Studio Digits software version 4.0 Li-Cor.

### 4.13. Quantification of Copper and Zinc in Bacteria Expressing UcMTs-GST

Recombinant bacteria were cultured in 100 mL of LB medium containing 100 mg mL^−1^ of carbenecillin until DO_600_ = 0.6, with 0.5 mM IPTG for 30 min, and with 1 mM CuSO_4_ or 1 mM ZnCl_2_ and IPTG for 6 h. Bacterial pellets showed a weight of 26–42 mg for copper cultures and 46–62 mg for zinc cultures. Pellets were dried at 60 °C for 48 h, suspended in 5 mL of 60% (*v*/*v*) HNO_3_, and incubated at 85 °C for 2 h. The solutions were filtered through 0.22 µm MCE filters (TCL, Santiago, Chile) and analyzed by flame atomic emission spectrophotometry using an atomic emission spectrophotometer ThermoFisher (Waltham, MA, USA).

### 4.14. Hierarchical Clustering of UcMTs

Amino acid sequences corresponding to MTs of different animal and plant species (234 in total) were selected from revised SwissProt repository of the UniprotKB database (www.uniprot.org). Alignment of these sequences was performed with Clustal W software with default setting. This alignment was used to generate the phylogenetic reconstruction to represent a hierarchical clustering using UPGMA algorithm based on distance. Phylogenetic and hierarchical clustering analyses were conducted using MEGA software version X [39].

### 4.15. Statistical Analyses

Statistical analyses were performed with the Prism 6 statistical package (Graph Pad software Inc., San Diego, CA, USA). Following confirmation of normality and homogeneity of variance, significant differences among treatments were determined by two-way ANOVA and Tukey’s multiple comparison post-hoc test, at a 95% confidence interval.

## Figures and Tables

**Figure 1 ijms-21-00153-f001:**
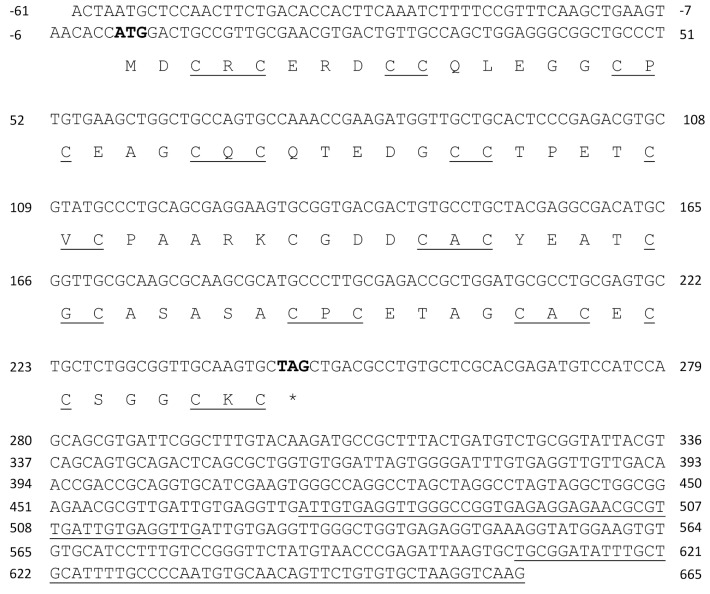
Complete cDNA and amino acid sequences of metallothionein UcMT1.1 (*Crassostrea*-like) from *U. compressa*. Initiation and termination codons are highlighted in bold; * indicate stop codon, cysteines arranged as CXC and CC motifs are underlined in the amino acid sequence of UcMT1. UcMT1.2 presents a deletion of 48 nucleotides located after the G in position 471 in the 3′UTR of MT1.1, and the deletion is underlined. UcMT3 present the deletion previously mentioned, and a deletion located after the C in position 607 in the 3′UTR of UcMT1.1, and the deletion is underlined.

**Figure 2 ijms-21-00153-f002:**
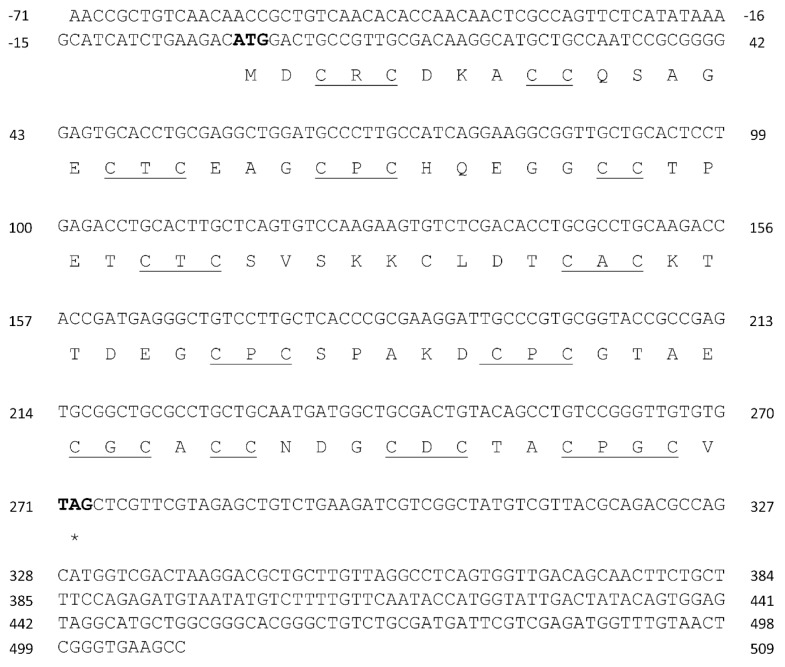
Complete cDNA and amino acid sequences of metallothionein UcMT2 (*Mytilus*-like) from the marine alga *U. compressa*. Initiation and termination codons are highlighted in bold; * indicate stop codon, cysteines arranged as CXC, CC, and CXXC motifs are underlined in the amino acid sequence of UcMT2.

**Figure 3 ijms-21-00153-f003:**
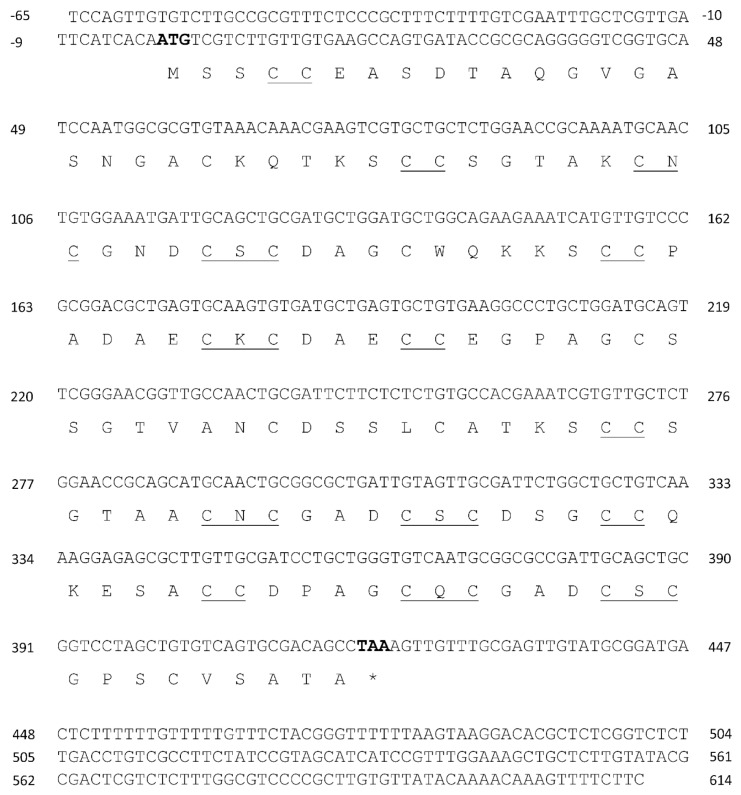
Complete cDNA and amino acid sequences of metallothionein UcMT3 (*Dreissena*-like) from the marine alga *U. compressa*. Initiation and termination codons are highlighted in bold; * indicate stop codon, cysteines arranged as CXC and CC motifs are underlined in the amino acid sequence of UcMT3.

**Figure 4 ijms-21-00153-f004:**
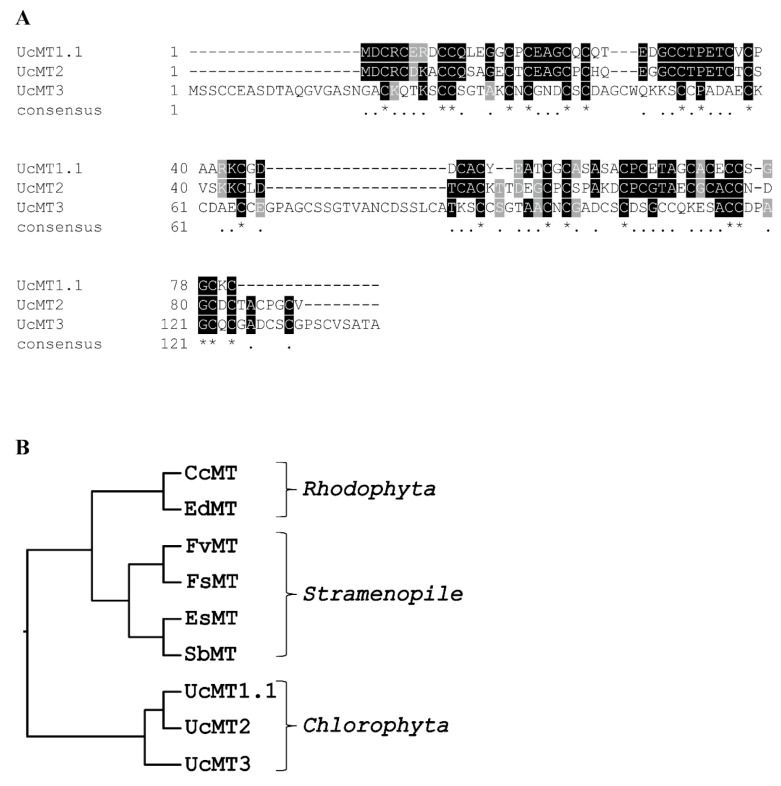
Alignment of amino acid sequences of metallotioneins (MTs) UcMT1.1, UcMT2, and UcMT3 from the marine alga *U. compressa* (**A**). Identical amino acids are indicated in black and similar amino acids are indicated in gray. * indicates identical amino acids and **·** similar amino acids. Hierarchical clustering of the amino acid sequences of UcMTs and MTs found in other marine macroalgae (**B**).

**Figure 5 ijms-21-00153-f005:**
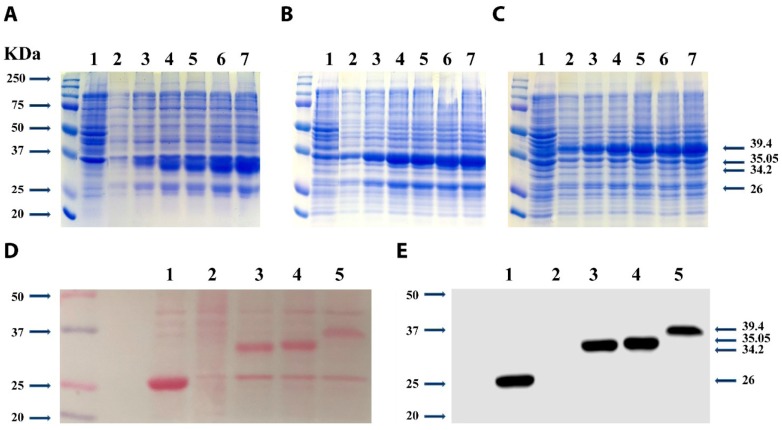
Visualization of protein extracts of bacteria overexpressing UcMT1.1-GST (**A**), UcMT2-GST (**B**), and UcMT3-GST (**C**) obtained from bacteria cultivated without IPTG (lane 1) or with 0.5 mM IPTG for 1, 3, 6, 9, and 12 h (lanes 2 to 7). Visualization of proteins bacterial extracts overexpressing UcMTs-GST transferred to a nitrocellulose membrane stained with Ponceau red dye (**D**) or incubated with anti-GST antibody and revealed by chemiluminescence (**E**). Proteins of extracts overexpressing GST (lane 1), not overexpressing GST (lane 2), and overexpressing UcMT1.1-GST (lane 3), UcMT2-GST (lane 4), and UcMT3-GST (lane 5) for 6 h. Arrows (left side) indicate molecular weights of standard proteins and arrows (right side) indicate molecular weights of UcMT3-GST (39.4 kDa), UcMT2-GST (35.05 kDa), UcMT1 (34.2 kDa), and GST (26 kDa).

**Figure 6 ijms-21-00153-f006:**
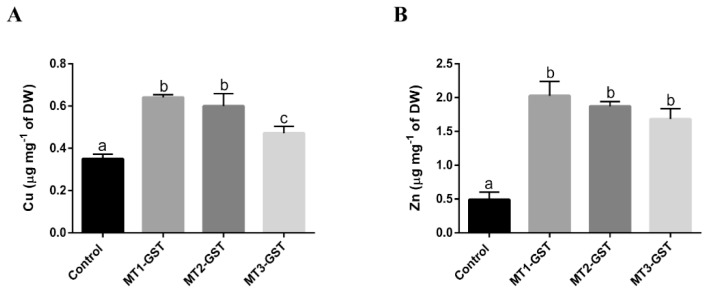
Level of copper (**A**) and zinc (**B**) in *E. coli* cultivated without IPTG for 6 h (control) and with 0.5 mM IPTG for 6 h and overexpressing UcMT1.1, UcMT2, and UcMT3. Levels of copper and zinc are expressed in micrograms per gram of bacterial dry weight (DW). Bars represent mean values of three independent experiments ± SD. Letters indicate significant differences among mean values (*p* < 0.05).

**Table 1 ijms-21-00153-t001:** Characteristics of *U. compressa* metallothioneins (UcMTs) transcripts and proteins. For transcripts: Nucleotides (nt); for proteins amino acids (aa) and cysteines (cys).

	Transcript (nt)	3′ UTR (nt)	ORF (nt)	5′ UTR (nt)	Protein (aa)	MW (kDa)	Total Cys	% Cys	Cys Motifs
**UcMT1.1**	726	61	246	418	81	8.2	25	29.4	3CC, 9CXC
**UcMT2**	580	71	273	236	90	9.05	27	30	3CC, 9CXC, 1CXXC
**UcMT3**	679	65	420	194	139	13.4	34	24	7CC, 7CXC

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
