# Peer review of "Isolation and Characterization of Copper- and Zinc- Binding Metallothioneins from the Marine Alga Ulva compressa (Chlorophyta)"

_ijms, 2019, doi:10.3390/ijms21010153_

Round 1

Reviewer 1 Report

I read the article "Isolation and characterization of copper- and zinc-binding metallothioneins of the marine alga Ulva
compressa (Chlorophyta)" by Zuniga et al. In the work the authors describe three metallothioneins from U. compressa, a green algae. While homologs from other species have been cloned and functionally assessed previously, little is known about these enzymes in green algae.

I find this article worthwhile to published. I had some difficulties reading the manuscript, because multiple track-changes made it hard to read. I have a few suggestions for improvement.

I think Fig S1 cold be included in the main text, but should be clearly described (Figure has no caption) and I could not find details how it was constructed. The introduction mainly deals with MTs in other species, so perhaps I would appreciate some sort of visualisation of the proteins and how they differ, this is very hard to follow in the text.

A second aspect I think that is noteworthy is the genomic features of U. compressa. While the authors perhaps chose a particular strain/clone - there is will be variation of the cloned genes within the species. I am not sure how this was addressed.

I find the analogy to other species and how closely related they are of interest. IS there perhaps knowledge about the metal concentrations other enzymes of other species can accumulate? Perhaps a comparative graph would help the reader about the specific features of U. compressa. Does U compressa also occur at locations outside Chile with high copper concentrations?

In summary, the article would benefit to be embedded in a more biology focused discussion, and also what the potential next steps could be after pinpointing how green alga might adapt to high metal concentrations.

Author Response

Answers to reviewer 1

1. I think Fig S1 could be included in the main text, but should be clearly described (Figure has no caption) and I could not find details how it was constructed.

We think it is difficult to incorporate Fig. S1 in the main text because the figure is too large and species names cannot be read without amplifying the figure. We added a paragraph in M&M to describe how the hierarchical clustering was performed. 

2. The introduction mainly deals with MTs in other species, so perhaps I would appreciate some sort of visualization of the proteins and how they differ, this is very hard to follow in the text.

A paragraph was added to explain this point 

3. While the authors perhaps chose a particular strain/clone - there is will be variation of the cloned genes within the species. I am not sure how this was addressed.
U. compressa was collected in the field and it does not correspond to a particular strain.

4. Is there perhaps knowledge about the metal concentrations other enzymes of other species can accumulate? Perhaps a comparative graph would help the reader about the specific features of U. compressa.

A paragraph was added to introduction in order to compare accumulation capacity of U. compressa in regard to other Ulva species.

5. Does U. compressa also occur at locations outside Chile with high copper concentrations?
U. compressa is a cosmopolitan species that accumulate heavy metals in different parts of the world. A sentence was added to introduction to clarify this point.

Reviewer 2 Report

The manuscript entitled “Isolation and characterization of copper- and zinc- binding metallothioneins of the marine alga Ulva compressa (Chlorophyta)” by Antonio Zúñiga, Daniel Laporte, Alberto González, Melissa Gómez, Claudio A. Sáez and Alejandra Moenne describes cloning and analysis of three metallothioneins from green marine alga. I found this manuscript very important in the field of research concerning metallothioneins since the research about metallothioneins in alga is underrepresent. This manuscript presents important and valuable data. However, I found several major and minor issues which needs to be addressed before the manuscript can be recommended for publication in IJMS.

Introduction

- page 2 lines 45-47 – the description of plant MTs is insufficient, four different types should be mentioned, and the explanation why linker region is 8 aa or 40 aa should be added (I think it is extremely important to provide comprehensive background of MTs from different phyla since analysis of the sequences of cloned MTs from U. compressa comprises a large part of the manuscript)

- page 2 lines 48 – the number of metal ions bound to a particular MT depends on the number of cysteine residues therefore it is completely wrong to claim that they bind 6-7 divalent or 11-12 monovalent cations

- page 2 line 51 – the first MT in plants was isolated from wheat germs by Lane and co-workers in 1987

- page 2 lines 50-59 – MTs are widespread throughout all kingdoms of living organisms therefore this part of the introduction should be rewritten to provide more comprehensive background to show how ubiquitous and diversified metallothioneins are

- page 2 line 57 – it is not clear what authors means by saying “In plants, there are six MTs….”

- the aim of the work should be presented more clearly

Results

- page 2 line 89 – should be were cloned or were amplified instead of were isolated

- page 2 line 91, page 3 lines 92-106 – this description of the features of cloned sequences is difficult to read, I would suggest presenting those data in the form of e.g. a table

- page 2 lines 95-98 – it is not clear how two other transcripts related to UcMT1.1 were isolated. Did authors confirmed that by RT-PCR to exclude that this a result of some errors in RACE PCR? It is very weird to have versions of transcript with deletion in UTRs.

- figure 1, 2, 3 – I am not sure whether it is necessary to present whole cDNA sequences in the text (maybe rather supplementary data)

- page 5 line 128, line 130 – linker as a description of Cys-free stretch between Cys-rich domains should be avoided. Linker suggest that it functions in formation of metal-binding clusters which was not confirmed for these MTs.

- page 6 line 149 – should be tag instead of tail

- page 7 line 173 – should be 83.71 instead of 83, 71. In addition, I would suggest to use the same number of decimal places. Authors gave only two values and should be three (for MT1, MT2 and MT3).

- page 8 line 176 – it is a bit confusing and do not allow easily to compare copper and zinc accumulation since here authors gave the results as x times more and for copper in %

- page 8 line 178-179 – I agree that the results have shown that bacteria overexpressed UvMTs accumulated more Cu and Zn but I do not see any supportive results for preference for zinc binding

- page 8 lines 185-206 – I found several major issues with this subchapter:

Why authors used different wavelengths? Ligand-to-metal charge transfer band is defined and for Zn-S is 230 nm and for Cu-S is broader and is around 280 nm. It is completely wrong to use different wavelengths.

Why so great excess of Cu+ or Zn2+ was added?

And the most important is what was the metal bound to a particluar MT at the beginning? Was it apo form? Or MT-metal form? How was it determined? Was this reaction titration of apo with metal or was it metal exchange? It seems that authors unfortunately completely forgot that MTs are metal bound protein and if metal was added to media during protein expression then probably protein was purified in MT-metal form (however this needs to be confirmed). And if this was Cu-MT then titration with zinc is nonsense since according to the Irving-Williams in vitro Cu(I) is bound to S more strongly than Zn and exchange reaction Cu-Zn cannot occur.

Discussion

- large part of the discussion is another description of the results

- discussion needs an extensive rewriting e.g. page 10 lines 249-262 – the whole paragraph is all over the same just in different words, no comments about the classification of MTs in Cu- and Zn-thioneins

- page 10 lines 226-248 – I found really surprising and interesting that MTs for this alga are rather similar to invertebrates than to other alga or plants. I think that authors should consider adding some alignments.

- page 10 line 240 – please do not use “structure” to describe sequence

- page 10 line 253-254 – I really cannot see evidences that zinc is preferentially bound by UvMTs

Materials and methods

- page 11 line 303 – Why Tris was used for washing? Tris can be a source of copper contamination.

- page 11 line 307 – should be 1 instead of one

- page 11 line 309, 311, 313 – should be 12,000 instead of 12.000

- page 12 line 333 – should be RNase H instead of RNAse H

- page 12-13 – I would suggest adding photos of the gels to supplementary data

- page 13 lines 384-389 -some details of this PCR reaction should be added

- page 13 line 392-393 – please delete

- page 13 line 407 – please add the company where DNA fragments where synthesised (of if this was in-house please add some details)

- page 13/14 lines 415-419 – I do not think that details about heat shock transformation of bacteria are needed

- page 14 lines 444-445 – Bradford method is not a proper method for determination of concertation of MTs

- page 15 line 465 – please add the concentration before ZnCl2

- page 15 line 465 – how was the concertation of CuSO4 and ZnCl2 determined? I think than 1 mM is much too high especially for copper and this had negative impact on the growth of bacteria (it was visible by the lower bacteria pellet mass grown in the presence of copper) and also on metabolism (and accumulation of metals). And also why copper was in the form of sulfate and zinc in the form of chloride (could have some impact on metabolism).

- page 15 line 469 – what were the wavelengths for Zn and Cu?

- page 15 lines 471-477 – the same questions I already asked in result section. And also why the reference cuvette contained Cu+ complex but not ZnCl2?

- page 15 lines 484, 502 – should be UV-Vis instead of UV-VIS

General comments

- the text needs moderate English language edition

- text need general editing and polishing (typos in the text and some in the references etc.)

Author Response

Answers to reviewer 2

Introduction

- page 2 lines 45-47 – the description of plant MTs is insufficient, four different types should be mentioned, and the explanation why linker region is 8 aa or 40 aa should be added (I think it is extremely important to provide comprehensive background of MTs from different phyla since analysis of the sequences of cloned MTs from U. compressa comprises a large part of the manuscript).

A paragraph was added describing the four types of plant MTs.

- page 2 lines 48 – the number of metal ions bound to a particular MT depends on the number of cysteine residues therefore it is completely wrong to claim that they bind 6-7 divalent or 11-12 monovalent cations.

This sentence was deleted.

- page 2 line 51 – the first MT in plants was isolated from wheat germs by Lane and co-workers in 1987

The reference was added.

- page 2 lines 50-59 – MTs are widespread throughout all kingdoms of living organisms therefore this part of the introduction should be rewritten to provide more comprehensive background to show how ubiquitous and diversified metallothioneins are.

A sentence mentioning the MT of the cyanobacteria Synechococcus was added and is highlighted in red.

- page 2 line 57 – it is not clear what authors means by saying “In plants, there are six MTs….”

The sentence was corrected

- the aim of the work should be presented more clearly

Done

- page 2 line 89 – should be were cloned or were amplified instead of were isolated

Corrected and highlighted in red.

- page 2 line 91, page 3 lines 92-106 – this description of the features of cloned sequences is difficult to read, I would suggest presenting those data in the form of e.g. a table

A table describing UcMT1.1, UcMT2 and UcMT3 was inserted in the text.

- page 2 lines 95-98 – it is not clear how two other transcripts related to UcMT1.1 were isolated. Did authors confirmed that by RT-PCR to exclude that this a result of some errors in RACE PCR? It is very weird to have versions of transcript with deletion in UTRs.

The amplification of 5´UTR of UcMT1 showed several times three bands that were isolated and sequenced. We think that there are several copies of UcMT1 in the genome of the alga U. compressa. A sentence was added in the discussion and is highlighted in red.

- figure 1, 2, 3 – I am not sure whether it is necessary to present whole cDNA sequences in the text (maybe rather supplementary data)

We prefer to present the whole sequence in the main text because you can have the whole picture of protein structures; you can easily see the position of aromatic residues, histidines and other important residues.

- page 5 line 128, line 130 – linker as a description of Cys-free stretch between Cys-rich domains should be avoided. Linker suggest that it functions in formation of metal-binding clusters which was not confirmed for these MTs.

Deleted

- page 6 line 149 – should be tag instead of tail

Changed and highlighted in red

- page 7 line 173 – should be 83.71 instead of 83, 71. In addition, I would suggest to use the same number of decimal places. Authors gave only two values and should be three (for MT1, MT2 and MT3).

Changed to 83.7 and the three values added as times of increase.

- page 8 line 176 – it is a bit confusing and do not allow easily to compare copper and zinc accumulation since here authors gave the results as x times more and for copper in %

We changed to times of increase.

- page 8 line 178-179 – I agree that the results have shown that bacteria overexpressed UvMTs accumulated more Cu and Zn but I do not see any supportive results for preference for zinc binding

We eliminated the sentence of preference

- page 8 lines 185-206 – I found several major issues with this subchapter:

Why authors used different wavelengths? Ligand-to-metal charge transfer band is defined and for Zn-S is 230 nm and for Cu-S is broader and is around 280 nm. It is completely wrong to use different wavelengths.

Figures were changed and they showed now only the shifts at 280 and 230 nm.

Why so great excess of Cu+ or Zn2+ was added?

Because Artrian and Capdevilla uses this range of concentrations.

And the most important is what was the metal bound to a particluar MT at the beginning? Was it apo form? Or MT-metal form? How was it determined?

Induction of MTs-GST in bacteria was performed without addition of metals. The LB medium contains a low amount of copper and zinc (0.1 mg/L for Cu and 0.4 mg/L for Zn). Thus, it is possible to assume that MTs-GST were mainly in its apo form.

Was this reaction titration of apo with metal or was it metal exchange? It seems that authors unfortunately completely forgot that MTs are metal bound protein and if metal was added to media during protein expression then probably protein was purified in MT-metal form (however this needs to be confirmed). And if this was Cu-MT then titration with zinc is nonsense since according to the Irving-Williams in vitro Cu(I) is bound to S more strongly than Zn and exchange reaction Cu-Zn cannot occur.

It was a titration with metals (see response above)

Discussion

large part of the discussion is another description of the results.

Redundancies were eliminated.

- discussion needs an extensive rewriting e.g. page 10 lines 249-262 – the whole paragraph is all over the same just in different words, no comments about the classification of MTs in Cu- and Zn-thioneins

Discussion about Cu and Zn thioneins is present in the discussion and is highlighted in red.

page 10 lines 226-248 – I found really surprising and interesting that MTs for this alga are rather similar to invertebrates than to other alga or plants. I think that authors should consider adding some alignments.

The MT of Fucus vesiculosus is also more similar to invertebrate MTs, so it is not that surprising. This point was added to discussion and is highlighted in red.

- page 10 line 240 – please do not use “structure” to describe sequence

Done and highlighted in red.

- page 10 line 253-254 – I really cannot see evidences that zinc is preferentially bound by UvMTs

We deleted this concept 

Materials and methods

- page 11 line 303 – Why Tris was used for washing? Tris can be a source of copper contamination.

The washing of the algae with Tris-EDTA is currently used to remove metals bound to cell wall that interfere with analytical procedures.

- page 11 line 307 – should be 1 instead of one

Changed and highlighted in red.

- page 11 line 309, 311, 313 – should be 12,000 instead of 12.000

changed

- page 12 line 333 – should be RNase H instead of RNAse H

Changed and highlighted in red.

- page 12-13 – I would suggest adding photos of the gels to supplementary data

We think this is not really necessary.

- page 13 lines 384-389 -some details of this PCR reaction should be added

Added and highlighted in red

- page 13 line 392-393 – please delete

deleted

- page 13 line 407 – please add the company where DNA fragments where synthesised (of if this was in-house please add some details)

Done and highlighted in red

- page 13/14 lines 415-419 – I do not think that details about heat shock transformation of bacteria are needed

Heat shock conditions were deleted.

- page 14 lines 444-445 – Bradford method is not a proper method for determination of concentration of MTs.

We did not have another method to perform MT quantification.

- page 15 line 465 – please add the concentration before ZnCl2

Done and highlighted in red

- page 15 line 465 – how was the concertation of CuSO4 and ZnCl2 determined? I think than 1 mM is much too high especially for copper and this had negative impact on the growth of bacteria (it was visible by the lower bacteria pellet mass grown in the presence of copper) and also on metabolism (and accumulation of metals). And also why copper was in the form of sulfate and zinc in the form of chloride (could have some impact on metabolism).

coli bacteria tolerate 1 mM copper or zinc in liquid culture and they grow without problem.

- page 15 line 469 – what were the wavelengths for Zn and Cu?

Now, we only showed the absorbance at 280 and 230 nm.

- page 15 lines 471-477 – the same questions I already asked in result section. And also why the reference cuvette contained Cu+ complex but not ZnCl2?

ZnCl2 was added and highlighted in red.

- page 15 lines 484, 502 – should be UV-Vis instead of UV-VIS

 Changed and highlighted in red

General comments

the text needs moderate English language edition done

- text need general editing and polishing (typos in the text and some in the references etc.)

Round 2

Reviewer 2 Report

The manuscript entitled “Isolation and characterization of metallothioneins 3 from the marine alga Ulva compressa (Chlorophyta)” by Antonio Zúñiga, Daniel Laporte, Alberto González, Melissa Gómez, Claudio A. Sáez and Alejandra Moenne is being reviewed second time. I appreciate corrections made by the authors according to some of my suggestions however in my opinion there are still some issues which needs to be further addressed.

1. Introduction

Some inaccuracies in the text:

- page 2 lines 42-43 – only in vertebrates

- page 2 line 56 – MT from wheat was not cloned but the protein was isolated

- page 2 line 59 – it is still not clear what authors means by “long linker of around 40 amino acids and a short linker of about 15 amino acids….”

- page 2 line 82 – please check the reference 26

2. Results

Paragraph 2.5

Even if any metal wasn’t added to the media during protein expression it cannot be assumed that the apo form was obtained during purification. From my experience even if I expressed MTs without extra metals added to the medium I obtained fully and partially metallated complexes. In fact, whether or not apo form was obtained needs to be determined by optical emission spectroscopy/mass spectrometry.

Metal exchanfe reactions should start from low concetration of metals (i.e. 1, 2, 3 etc. equivalets with respect to protein concetration).

Apo form of MT is prone to oxidation therefore it has to be protected by at least using nitrogen or argon purged buffers.

The concertation of MTs is wrong since Bradford does not give proper results for metallothioneins.

Therefore, I do not know what authors would like to show by this experiment. The ability to bind Zn and Cu by cloned MTs is already shown by enhanced accumulation of Cu and Zn in bacteria cells. Experiment present in this paragraph is not designed properly and in my opinion without determination of the nature of MTs purified from bacterial cells (apo or metallated complex and what metal) and proper determination of concentration of MT cannot be accepted.

3. General

Still some polishing of the text is needed.

Author Response

New corrections of reviewer 2 were incorporated. The last figure was deleted as well as conclusions derived from this figure. We thank reviewer 2 for information provided. Prof. Alejandra Moenne.

Round 3

Reviewer 2 Report

I found this paper interesting and important in the reasearch field of metallothoneins.